# PLGA Nanoparticle-Based Dissolving Microneedle Vaccine of *Clostridium perfringens* ε Toxin

**DOI:** 10.3390/toxins15070461

**Published:** 2023-07-19

**Authors:** Wei Wan, Yue Li, Jing Wang, Zhiying Jin, Wenwen Xin, Lin Kang, Junhong Wang, Xiaoyang Li, Yakun Cao, Hao Yang, Jinglin Wang, Shan Gao

**Affiliations:** 1State Key Laboratory of Pathogen and Biosecurity, Institute of Microbiology and Epidemiology, Academy of Military Medical Sciences (AMMS), Beijing 100071, China; wanwan03012022@163.com (W.W.); lyzaokewenhua@163.com (Y.L.); amms_wj@163.com (J.W.); jinzhiying96@163.com (Z.J.); xinww@hotmail.com (W.X.); kang_lin@hotmail.com (L.K.); lxy2563922462@163.com (X.L.); caoyakun0604@163.com (Y.C.); 2College of Life Science and Technology, Beijing University of Chemical Technology, Beijing 100029, China; 3Hebei Key Laboratory of Animal Physiology, Biochemistry and Molecular Biology, College of Life Sciences, Hebei Normal University, Shijiazhuang 050024, China; 4School of Medical Technology, Tianjin University of Traditional Chinese Medicine, Tianjin 301617, China; wangjunhong11007@163.com; 5Beijing Noninvasion Biomedical Technology Co., Ltd., Beijing 101111, China; iam@noninvasion.com

**Keywords:** *Clostridium perfringens* epsilon toxin, dissolving microneedle patch, poly(lactic-co-glycolic acid, vaccine adjuvants

## Abstract

Epsilon toxin (ETX) is an exotoxin produced by type B and D *Clostridium perfringens* that causes enterotoxemia or necrotic enteritis in animals such as goats, sheep, and cattle. Vaccination is a key method in preventing such diseases. In this study, we developed a new type of dissolving microneedle patch (dMN) with a nanoparticle adjuvant for enhanced immune response to deliver the rETX^Y196E^-C protein vaccine. We chose FDA-approved poly(lactic-co-glycolic acid) (PLGA) to prepare nanospheres as the vaccine adjuvant and introduced dimethyldioctadecylammonium bromide (DDAB) to make the surface of PLGA nanoparticles (PLGA NPs) positively charged for antigen adsorption. PLGA NPs with a diameter of 100~200 nm, a surface ZETA potential of approximately +40 mV, and good safety were successfully prepared and could effectively adsorb rETX^Y196E^-C protein. Using non-toxic and antibacterial fish gelatin as the microneedle (MN) matrix, we prepared a PLGA-DDAB dMN vaccine with good mechanical properties that successfully penetrated the skin. After immunization of subcutaneous (SC) and dMN, antibody titers of the PLGA and Al adjuvant groups were similar in both two immune ways. However, in vivo neutralization experiments showed that the dMN vaccines had a better protective effect. When challenged with 100 × LD_50_ GST-ETX, the survival rate of the MN group was 100%, while that of the SC Al group was 80%. However, a 100% protective effect was achieved in both immunization methods using PLGA NPs. In vitro neutralization experiments showed that the serum antibodies from the dMN and SC PLGA NPs groups both protect naive mice from 10 × LD_50_ GST-ETX attack after being diluted 20 times and could also protect MDCK cells from 20 × CT_50_ GST-ETX attack. In conclusion, the PLGA-DDAB dMN vaccine we prepared has good mechanical properties, immunogenicity, and protection, and can effectively prevent ETX poisoning. This provides a better way of delivering protein vaccines.

## 1. Introduction

Epsilon toxin (ETX) is a pathogenic factor produced by *Clostridium perfringens* types B and D that can cause enterotoxemia or necrotizing enteritis in lambs and calves. Its toxicity is second only to that of botulinum and tetanus toxins, and it is classified as an important biological terrorist agent by the international community. ETX has the characteristics of rapid onset, short course, and high mortality rate. It is difficult to achieve a therapeutic effect by medication alone after poisoning, and acute cases often result in death before medication can take effect, causing huge losses to the livestock industry. The potential hazards of ETX to humans and animals, as well as the threat of bioterrorist attacks worldwide, have accelerated the development of vaccines to significantly reduce the incidence and mortality of ETX poisoning [1].

Most traditional vaccines used to prevent ETX poisoning are detoxified protein vaccines that can cause inflammation and damage to animals, reducing their safety. Compared with traditional attenuated vaccines, recombinant protein vaccines have significant advantages such as convenient production, fast speed, simple purification, high yield, small side effects, and higher safety, and have gradually replaced traditional detoxified vaccines. However, due to their low molecular weight, poor immunogenicity, and rapid vaccine release, they cannot stimulate the production of long-lasting and effective immune responses [2]; aluminum (Al) adjuvants are typically needed to achieve the expected immune effect. Al adjuvants are widely used and universal vaccine adjuvants that can promote both Th2 humoral immune responses and cell-mediated immune responses [3]. However, they have side effects such as neurotoxicity, local fever, and swelling after injection; thus, an urgent need for safe and effective new adjuvants exists [4,5]. Recently, biodegradable polymer particle adjuvants have received extensive attention, especially poly(lactic-co-glycolic acid) nanoparticles (PLGA NPs), which are FDA-approved pharmaceutical excipients [6] with high safety and biodegradability. PLGA NPs can adsorb or encapsulate antigens, acting as vaccine carriers, forming a vaccine storage depot, delivering antigens and prolonging their activity time. They can also enhance antigen uptake, activation, and maturation of dendritic cells (DC) [7], which can enhance cellular immunity as an immunological adjuvant. Researchers have used PLGA as an adjuvant to deliver protein antigens such as ovalbumin (OVA) [7,8], hepatitis B [9,10], malaria [11,12], and *Chlamydia trachomatis* [13,14], all of which have effectively and safely induced immune responses. 

As a new vaccine delivery approach, microneedles (MNs) penetrate the stratum corneum with their micron-sized needle-like projections, delivering vaccines to Langerhans cells and dendritic cells in the epidermis and inducing an adaptive immune response [15,16]. Compared to traditional vaccination methods, MN vaccines have many advantages, including good heat stability, self-administration, almost painless administration, and no sharps waste [17,18,19]. By combining PLGA NPs containing antigens with MN delivery technology, a new type of immuno-enhanced MN is explored. Using PLGA NPs to protect antigens from environmental influences and enhance the permeability of nanoparticles in the skin [20], the long-term stability of antigens can be improved [21] and the sustained release of antigens can be extended [7] to enhance the immune effects of vaccine antigens. The ease and convenience of MN delivery make possible the self-administration of vaccines, thereby improving vaccination efficiency. By combining the advantages of the above two technologies, the preventive effect of vaccines can be enhanced, providing an effective means for preventing ETX toxicity. Zheng and co-workers have reported a successful case of combining PLGA NPs vaccines with MN to deliver hepatitis B protein vaccines to the skin [10].

In this study, we develop a dissolving microneedle patch (dMN) using a PLGA NPs adjuvant to deliver the rETX^Y196E^-C protein vaccine. We introduce the addition of dimethyldioctadecylammonium bromide (DDAB) to make the surface of PLGA NPs positively charged for antigen adsorption, creating a PLGA-DDAB/rETX^Y196E^-C dMN vaccine with significant advantages, including safety, high efficiency, and easy administration. The sustained-release effect of PLGA NPs as an adjuvant was evaluated and the immune response type and intensity stimulated by the dMN vaccine in mice were detected and analyzed. This effectively improved the problems of weak immunogenicity and inconvenient injection of traditional ETX recombinant protein vaccines.

## 2. Results

### 2.1. Preparation of Vaccine Antigens and Toxin Proteins

We prepared attenuated mutant protein rETX^Y196E^-C and GST-ETX recombinant protein using bacterial strains stored in our laboratory. The rETX^Y196E^-C protein was obtained by substituting the 196th tyrosine (Y196) with glutamic acid (E) and introducing a C-terminal peptide consisting of 23 amino acids, which is a variant with low toxicity used for mouse immunization, while the GST-ETX protein retained toxicity and was used for immunization and subsequent challenge. We purified the proteins using nickel-nitrilotriacetic affinity chromatography and gradient elution. The purified rETX^Y196E^-C protein had a molecular weight of 33 kDa (Figure 1A,B), while the purified GST-ETX protein had a molecular weight of 56 kDa (Figure 1C,D). The purity of both proteins was greater than 81% as determined by SDS-PAGE analysis. The concentrated protein was used for subsequent experiments. The concentration of rETX^Y196E^-C and GST-ETX proteins were estimated as 31.08 mg/mL and 2.12 mg/mL, respectively, using a BCA protein quantification kit.

### 2.2. Characterization of PLGA-DDAB Nanoparticles

The surface of simple PLGA NPs is negatively charged and cannot directly adsorb negatively charged proteins. However, the introduction of a cationic lipid, DDAB, makes the surface of PLGA NPs positively charged, allowing the negatively charged antigen (Ag), rETX^Y196E^-C protein, to be adsorbed onto their surface through electrostatic interactions (Figure 2A). PLGA-DDAB NPs were successfully prepared using the emulsion–evaporation method. SEM scanning electron microscopy (Figure 2B) and field emission transmission electron microscopy (Figure 2C) showed that the PLGA-DDAB NPs were spherical, smooth-surfaced, had a particle size of about 100 nm, and could be efficiently absorbed by antigen-presenting cells (APCs) to enhance immune responses [22]. The introduction of DDAB increased the particle size of PLGA NPs by 12.71 ± 2.895 nm, indicating the successful introduction of DDAB into PLGA NPs (Figure 2E). To further verify this result, EDS spectral scanning analysis of the Br and N elements unique to DDAB was used to confirm the presence of both elements in PLGA NPs (Figure 2F,H). The amount of DDAB added significantly affected the ZETA potential of the PLGA surface (Figure 2D), which in turn affected the protein adsorption efficiency. Therefore, we added 0.005, 0.01, 0.02, or 0.03 g of DDAB to a 0.1 g PLGA solution and observed the effect of different amounts of DDAB on the ZETA potential of the particle surface. When 0.02 g of DDAB was added, the ZETA potential of PLGA-DDAB NPs reached the highest level. The ZETA potential and polymer dispersity index (PDI) can directly reflect the stability and dispersibility of the particles. The particle size distribution, ZETA potential, and PDI of the NPs were determined using a laser particle size analyzer. Each sample was measured three times and the mean value was used in the analyses (Table 1). The PDI of PLGA-DDAB was less than 0.1, indicating good stability and dispersibility of the particles, and full adsorption of protein antigens.

We evaluated the adsorption efficiency and safety of the prepared PLGA-DDAB NPs to ensure the nanoparticles could be used in subsequent mouse immunization experiments. The concentration of rETX^Y196E^-C protein in the supernatant before and after adsorption by PLGA-DDAB NPs was measured using the bicinchoninic acid (BCA) assay method and the adsorption efficiency of PLGA-DDAB NPs to rETX^Y196E^-C protein calculated at approximately 60%. The cytotoxicity of the nanoparticles was measured by co-culturing Madin–Darby canine kidney (MDCK) cells with PLGA NPs and PLGA-DDAB NPs. The survival rate of MDCKS cells was approximately 90% for both types of nanoparticles (Figure 2G), indicating good biocompatibility of PLGA-DDAB NPs for subsequent vaccine delivery.

### 2.3. Analysis of the Performance of PLGA-DDAB dMN

We used fish gelatin and sucrose as MN matrices and prepared PLGA-DDAB dissolving microneedle patches (dMN) using a mold (Figure 3A). We used a two-step method to prepare the MN tips and backings separately to ensure that the vaccine was concentrated in the tip of the needle. The unmolded MN patch was placed under a stereomicroscope for observation; Figure 3B shows that the 14 × 14 MN array was complete and the tips were intact.

We used PLGA fluorescent NPs encapsulating rhodamine B to verify whether the dMN prepared in this experiment concentrated the drug at the tip of the needle. MNs were made using the same preparation method, and the position of the fluorescent nanoparticles was observed using a fluorescence inverted microscope. The PLGA fluorescent NPs were mainly concentrated at the tip of the needle (Figure 3F), indicating that the antigen was concentrated at the tip of the MN and could achieve accurate and quantitative delivery. The PLGA-DDAB/rETX^Y196E^-C MN was quantified using a sandwich Elisa method, and the average content of rETX^Y196E^-C in the PLGA-DDAB dMN was calculated to be 9.86 ± 2.47 μg/patch, which was equivalent to that of the needle group.

To evaluate the ability of the PLGA-DDAB dMN to penetrate the skin, mechanical performance evaluation was carried out using a force-displacement testing machine. As shown in Figure 3C, after the force gauge probe was moved 0.3 mm towards the backing of the patch with an axial pressure of 20 N, only the tip of the MN was slightly deformed (Figure 3E), indicating that PLGA-DDAB dMN can withstand at least 20 N of pressure and has sufficient mechanical strength to pierce the mouse epidermis layer and deliver the antigen subcutaneously. Optical coherence tomography (OCT) can reveal significant breaks in the epidermal layer of pig skin caused by the penetration of microneedles. As the dMN gradually dissolves within the skin, these breaks also disappear in the OCT images. This phenomenon can reveal the dissolution time of the dMN after piercing the skin. The dMNs had mostly dissolved in pig skin 10–15 min after piercing (Figure 3G), indicating that the dMN had good biocompatibility and could dissolve into the skin in a relatively short time.

### 2.4. In Vivo Release Time and Antibody Titer Analysis of MN Vaccines

The indirect ELISA method was used to estimate the serum titers of mice immunized with subcutaneous (SC) or dMN immunizations in each of three groups: Ag, Ag + Al, and Ag + PLGA groups. The SC Ag group had a low antibody titer, but the other five groups all produced antibody titers above 10^4^ (Figure 4C). Although the antibody titers of the Ag + PLGA groups were higher than those of the Ag and Ag + Al groups, the difference was not statistically significant. Only the antibody titer of the SC Ag + PLGA group showed a statistically significant difference from that of the SC Ag group after the second and third immunizations. Therefore, we believe that both the Ag + Al and Ag + PLGA groups of SC immunization and dMN immunization can produce strong immune effects.

To analyze whether Al adjuvant and PLGA NPs adjuvant have similar immune-enhancing mechanisms, a mouse in vivo imaging experiment was performed to compare the antigen release time in vivo. The rETX^Y196E^-C protein labeled with Cy5.5 dye was delivered using the same immunization methods as above, and the fluorescent signal was collected using the IVIS Spectrum imaging system (PerkinElmer, Waltham, MA, USA). Interestingly, we found that there was no correspondence between the antigen release time and the antibody titer. Both SC and dMN immunizations with Ag groups showed no fluorescence signal after 2 d post-vaccine (Figure 4A), yet the antibody titer of the dMN Ag group was higher (Figure 4C). This result shows that the use of dMN has the potential to improve immune responses. The dMN Ag + Al group stopped showing an antigen fluorescence signal more than 2 days before the SC group. Similarly, the dMN Ag + PLGA group stopped showing a signal more than 5 d before the SC group (Figure 4A), but the antibody titer between PLGA and AI groups did not differ and was high in both groups (Figure 4C). The prolonged antigen release time of the AI immunization group is indicative of the sustained release effect of the aluminum adjuvant, which enhances the immune response [23]. No sustained release phenomenon was observed in either Ag + PLGA group, and thus, the immune-enhancing mechanisms of the two adjuvants clearly differ. We speculate that the Ag + PLGA immune-enhancing mechanism may be related to its easier uptake by antigen-presenting cells, but further exploration of this mechanism is needed.

### 2.5. Analysis of Immune Response Types

To evaluate whether Ag + PLGA can effectively promote Th1 and Th2 immune responses, we estimated the type of immune response induced by different adjuvants by measuring the IgG subtypes (IgG2a and IgG1) in mouse serum and cytokines in splenocytes after immunization. The ratio of IgG2a to IgG1 and cytokines represents the polarization effect of the immune response in both Th1 and Th2 aspects. Unlike aluminum adjuvant, which tends to stimulate Th2 immune response, PLGA NPs generated a higher ratio of IgG2a to IgG1 (Figure 5A) and significantly promoted production of IL-2 and TNF-γ (Figure 5D,E), indicating an induced Th1 immune response, and activation of CD4^+^ T and CD8^+^ T cells (Figure 5B,C), further demonstrating that Ag + PLGA effectively enhances cellular immunity. In addition, the IL-4 and IL-6 induced by Ag + PLGA were both higher than those induced by the aluminum adjuvant (Figure 5F,G), indicating that they also enhance humoral immunity. Based on these results, dMN delivery can achieve the same immune effect as subcutaneous injection, and Ag + PLGA can effectively enhance immunogenicity by stimulating the production of two types of immune response.

### 2.6. Protective Effect of Ag + PLGA dMN Vaccine

The protective effect of the dMN vaccine on mice was explored through in vivo neutralization experiments, in vitro neutralization experiments, and pathological analysis.

In the in vivo neutralization experiment, immunized mice were intraperitoneally injected with 100 × LD_50_ GST-ETX toxin and observed for 3 d. All mice in the PBS control group died within 6 h; two out of three subcutaneously immunized groups died, including all mice in the SC immunization Ag group within 24 h; one mouse in the SC immunization Ag + Al group died during the 72 h observation period, while all mice in the SC immunization Ag + PLGA group survived (Figure 6A). All three groups of mice immunized with dMN containing the same antigenic components survived (Figure 6A); even the dMN Ag group without any adjuvant was able to resist the attack of 100 × LD_50_ GST-ETX toxin.

The in vitro neutralization experiment included both animal and cell experiments. The immune sera of each of the six groups of mice were mixed with GST-ETX toxin, incubated at 37 °C for 30 min, and then injected into blank mice and co-cultured with cells to observe the survival of mice and cells. Both mouse survival (Figure 6B) and cell proliferation curves (Figure 6D) showed that the immune sera of the SC immunization Ag group did not protect either mice or cells, and the immune sera of the SC immunization Ag + Al group protected mice but did not effectively protect cells. The immune sera of the SC immunization Ag + PLGA group and the three groups of mice immunized with dMN were able to protect naive mice from 10 × LD_50_ GST-ETX toxin after a 20-fold dilution and protect cells from the attack of 20 × CT_50_ GST-ETX toxin (see Figure 6C). These results are consistent with those of the in vivo neutralization experiment.

For pathological analysis, we used HE staining to observe the kidney and brain of mice at death or 7 d after toxin injection. Positive control mice were intraperitoneally injected with 100 × LD_50_; intraperitoneal injection of PBS was used as a negative control. Positive control mice lost mobility within 10 min of injection and were euthanized immediately when they were close to death. Pathological section results (Figure 6E) showed that the positive control mice had significant bleeding and edema in the kidney, and significant vacuolization, bleeding, and edema in the brain, while there were no pathological changes in the kidney and brain tissue of the negative control and immunized mice, except for the SC immunization Ag group. These results indicate that all immunization groups except for the SC immunization Ag group were protected from the toxin.

## 3. Discussion

A dMN containing PLGA NPs was successfully prepared for the delivery of the rETX^Y196E^-C protein vaccine in this study. The patch has advantages, such as enhanced immune response, quantitative delivery, and significantly reduced pain, and effectively protected mice against the toxin at 100 × LD_50_ GST-ETX.

PLGA is an FDA-approved biodegradable polymer with good biocompatibility, and microspheres prepared using PLGA as an adjuvant or delivery system have gained wide attention. PLGA can interact with antigen-presenting cells to induce an ideal immune response and has been used as an adjuvant in vaccines against various diseases, including hepatitis B [9,10], malaria [11,12], and tumors [24]. PLGA has also been shown to increase antibody titers to some extent. However, there are also certain problems in the application process, such as the complex and cumbersome coupling process between the antigen and PLGA NPs, which can lead to antigen inactivation, the toxicity of crosslinking or coupling agents that render the vaccine unusable, and prolonged antigen release from PLGA NPs leading to immune tolerance. In our study, cationic lipid DDAB was introduced into the PLGA NPs we prepared to give them a positive surface charge, allowing a gentle and efficient loading of the mutant protein vaccine rETX^Y196E^ -C prepared in our laboratory, reducing the impact of the coupling process on antigen activity.

In the evaluation of immune effects, we found that, although both aluminum adjuvant and PLGA NPs can stimulate mice to increase antibody titers, their mechanisms of enhancement differ, with aluminum adjuvant having a longer release time. Although the PLGA NPs’ release time is somewhat extended, it is significantly weaker than for the aluminum adjuvant. Therefore, we speculate that the immune enhancement mechanism of PLGA NPs may be related to their easier uptake by antigen-presenting cells. Immunoassay results indicate that PLGA NP can be used as an alternative to aluminum adjuvant. In the neutralization experiments, the protective effect of dMN immunization on mice was superior to subcutaneous immunization, consistent with previous reports [25,26,27], showing that MN immunization also has an immune enhancement effect. Therefore, the combined use of PLGA NPs and dMN immunization can effectively enhance the immune effect of vaccine antigens.

The issue of accurate quantitative delivery of vaccine proteins in dMN remains a challenge, impeding the application of such vaccines. Various research teams are attempting to solve this problem. For example, the Prausnitz team has used different polymers to prepare the tip and backing of the MN, concentrating the drug into the tip and thereby avoiding diffusion to the backing [28]. They also employed freeze-drying to prepare the tip for quick separation from the backing, enabling complete delivery of the vaccine antigen into the dermis. Several Chinese teams, including Xin Dong Guo and Chenjie Xu, have improved the preparation process to form a bubble cavity between the tip and the backing, facilitating complete drug delivery into the skin [29,30,31,32]. In this study, we employed centrifugation during the MN preparation process to concentrate the PLGA NPs adsorbed with vaccine proteins at the tip of the MN, thereby preventing diffusion to the backing and achieving quantitative delivery of vaccine antigens to the dermis.

This dMN vaccine patch has the benefits of quick and easy administration, painlessness, and excellent thermal stability, which are characteristic of dMN vaccine patches. In addition, it is enhanced with PLGA NPs to provide a more robust immune response. Furthermore, the preparation method is simple, safe, effective, and easily scalable for mass production. This patch provides an effective means of preventing ETX enterotoxemia, reducing losses in animal husbandry resulting from ETX toxicity, and guarding against the biological security threat posed by ETX as a bioterrorism agent.

## 4. Conclusions

We have successfully prepared PLGA NPs with a smooth surface, uniform size, good safety, and positive charge that can gently and efficiently adsorb rETX^Y196E^-C vaccine antigens. These nanoparticles are combined with dMN to deliver antigens. We validated the reliability of the Ag + PLGA dMN vaccine patch in terms of mechanical properties, skin penetration, immunogenicity, and protection. We show that the patch had excellent mechanical properties and can withstand a pressure of 20 N without significant deformation while successfully penetrating the skin and achieving transdermal vaccine delivery. The vaccine enhanced both cellular and humoral immune responses, with a specific IgG antibody titer of 10^5^ after three immunizations, and effectively activated T cells and the expression of cytokines such as IL-2, IL-4, IL-6, and interferon-gamma. This approach provides a new powerful tool for preventing ETX intoxication and bioterrorism attacks and has the potential to become a new vaccine delivery system for other vaccines or drugs.

## 5. Materials and Methods

### 5.1. Animals

SPF-grade female Balb/c mice (6–8 weeks old) were purchased from SPF (Beijing, China) Biotechnology Co., Ltd. The experimental mice were housed in the SPF animal room of the Experimental Animal Center of the Academy of Military Medical Sciences, and the animal experiments were reviewed and approved by the Animal Ethics Committee of the Academy of Military Medical Sciences. All animal experiments complied with the requirements of the Guidelines for Ethical Review of Laboratory Animal Welfare of the Academy of Military Medical Sciences.

### 5.2. Preparation of Vaccine Antigens and Toxin Proteins

The rETX^Y196E^-C and GST-ETX recombinant strains were stored in our laboratory, and detailed protein preparation methods followed that of previous publications [32,33,34]. Briefly, the bacterial strains were resuscitated in LB medium and, after expansion culture, the bacterial cells were disrupted using ultrasonic treatment. The supernatant containing the target protein was obtained by centrifugation at 11,000× *g* for 20 min. Nickel-nitrilotriacetic (GE Healthcare, Piscataway, NJ, USA) and glutathione Sepharose (GE Healthcare, Piscataway, NJ, USA) were used to perform affinity chromatography on the rETX^Y196E^-C and GST-ETX recombinant proteins, respectively. The purified products were concentrated using a 10 kDa ultrafiltration tube, and imidazole in the solution was displaced from the system by repeating the process three times. The molecular weight and purity of the purified products were confirmed by SDS-PAGE gel electrophoresis. A BCA protein quantitation kit (PA101-01, Biomed, Beijing, China) was used to determine protein concentration. 

### 5.3. Preparation and Characterization of PLGA-DDAB Nanoparticles

PLGA-DDAB NPs were prepared using an emulsion evaporation method [6,35]. Briefly, PLGA (50:50 poly (DL-Lactide-co-glycolide), carboxylate end group (nominal); LACTEL Absorbable Polymers, Birmingham, AL, USA) and DDAB (Sigma-Aldrich, St. Louis, MO, USA) were dissolved in 10 mL of DCM (Sinopharm Chemical Reagent Co., Ltd., Shanghai, China) was used as the oil phase and 0.5 g of PVA (Sigma-Aldrich, St. Louis, MO, USA) was dissolved in 50 mL of ddH_2_O as the aqueous phase. The oil phase was slowly added to the aqueous phase at 600 rpm and emulsified for 30 min (125 W, 75%) using an ultrasonic disintegrator (Q125; Qsonica LLC, Newton, CT, USA). The large pellets were precipitated by centrifugation at 4000× *g* for 10 min, the supernatant was resuspended in ddH_2_O and washed three times by centrifugation at 20,000× *g* for 30 min to obtain PLGA-DDAB NPs, which were resuspended in 2 mL of ddH_2_O with 3% sucrose (V900116, Sigma, St. Louis, MO, USA) as lyophilization protectant. 

The same method was used for the preparation of fluorescent PLGA-DDAB NPs. Rhodamine B was dissolved in dd H_2_O as the internal aqueous phase, PLGA and DDAB were dissolved in DCM as the oil phase, and the internal aqueous phase was added to the oil phase in a ratio of 1:5 internal aqueous phase:oil phase, sonicated for 2 min (125 W, 75%), and then added to 1% PVA (*w*/*v*) external aqueous phase solution and sonicated for 30 min (125 W, 75%). Water-in-oil-in-water emulsions were obtained; the organic solvent was evaporated by stirring overnight in a fume hood protected from light, ddH_2_O-washed three times and then lyophilized to obtain fluorescent PLGA-DDAB NPs.

PLGA-DDAB NP surface morphology was observed using a field emission scanning electron microscope (SU8020, Hitachi Limited, Tokyo, Japan) and a field emission transmission electron microscope (FEI Talos F200X, FEI Company, Hillsboro, OR, USA). The particle size distribution, dispersibility index (PDI), and ZETA potential were determined using a laser particle sizer (Zetasizer Nano ZS90, Malvern Instruments Ltd., Malvern, UK). Energy dispersive X-ray spectroscopy (EDX) was used to analyze the presence of Br and N elements in the DDAB within the PLGA nanoparticles.

### 5.4. Adsorption Rate Testing

To detect the adsorption rate of rETX^Y196E^-C protein onto PLGA-DDAB NPs, the following series of steps was performed. PLGA-DDAB NPs in freeze-dried powder form were reconstituted in 500 μL of ultrapure water. The mixture was then sonicated to achieve dispersion. A predetermined volume of rETX^Y196E^-C protein solution at a specific concentration was added to the PLGA-DDAB NPs dispersion. The solution was then gently mixed at 4 °C for a defined period to allow for the formation of PLGA-DDAB/rETX^Y196E^-C nanoparticle suspension. Subsequently, the suspension was centrifuged at 20,000× *g* for 3 min and the supernatant was collected. The concentration of rETX^Y196E^-C protein in the supernatant was estimated using the bicinchoninic acid (BCA) assay method. The amount of rETX^Y196E^-C protein adsorbed onto PLGA-DDAB nanoparticles was calculated by subtracting the concentration of rETX^Y196E^-C protein in the supernatant after adsorption from the initial concentration of added antigen.

### 5.5. Cell Viability Assay for PLGA-DDAB Nanoparticles

MDCK cells were seeded in 96-well plates at a density of 1 × 10^5^ cells/mL and incubated overnight at 37 °C with 5% CO_2_. PLGA-DDAB NPs were diluted in DMEM containing 10% FBS at different concentrations and added to 96-well plates. The plates were incubated for 1 h at 37 °C with 5% CO_2_, then the medium was removed and the cells were washed with PBS three times. Fresh medium (100 μL) and MTS (20 μL) were added to each well and the plates were incubated under the same conditions for 3 h. The absorbance at 492 nm was measured using a microplate reader (Multiskan Mk3, Thermo Fisher, Waltham, MA, USA), and cell viability was calculated accordingly.

### 5.6. Preparation and Characterization of PLGA-DDAB dMN

The PLGA-DDAB dMN was prepared using a two-step method and a PDMS mold provided by Shanghai Jiao Tong University [36]. The MNs were conical in shape with a height of 650 μm, a bottom diameter of 360 μm, and a spacing of 360 μm [37]. In the first step, the needle tip of the MN was prepared. PLGA-DDAB containing rETX^Y196E^-C protein or blank PLGA-DDAB was suspended in a solution of 35% gelatin (G7041, Sigma, USA) (m/v) and 10% sucrose (m/v) in water. The solution was centrifuged at 3500 rpm for 10 min (Beckman Coulter, 21R Centrifuge, San Pablo, CA, USA) to allow the solution to completely enter the needle tip. Any excess solution was then removed. In the second step, the backing of the MN was prepared. A water solution of the same matrix was added, and the mold was centrifuged at 3500 rpm for 10 min. The mold was then removed and placed in a drying oven for 24 h. The MN patch was carefully removed with forceps and sealed for storage.

### 5.7. MN Performance Assessment

Mechanical properties of the PLGA-DDAB dMN was evaluated using a force-displacement testing machine (MARK-10, Copiague, NY, USA). The MNs were placed horizontally on the platform of the testing machine with the needle tips facing upward. The force gauge probe was set to move downwards at a uniform speed of 0.1 mm/s until the maximum force of 20 N·m was reached. The force-displacement curves were analyzed, and the needle tip deformation was observed using an optical microscope (Olympus DP71, Tokyo, Japan) after force application.

The PLGA-DDAB dMN was tested for skin penetration using pig skin and optical coherence tomography (MDL, VivoSight DX, Kent, UK). The MNs were placed on the pig skin, and pressure was applied using the thumb for 0, 5, 10, and 15 min. The patches were then left on the skin, and OCT scanning was performed immediately to visualize the skin penetration of the MN.

### 5.8. In Vivo Release Time Determination

The experimental mice were divided into two immunization (SC, MN) by three vaccine-type (Ag, Ag + Al, Ag + PLGA) groups for six groups total: SC immunization with antigen group, SC immunization with Ag + Al group, SC immunization with Ag + PLGA group, MN Ag group, MN Ag + Al group, and MN Ag + PLGA group. Prior to administration, the back hair of mice was removed with depilatory cream and then wiped with saline and dried with a dry cotton ball. The doses of each group were kept the same, and the subcutaneous needle injection group was injected subcutaneously at two points on the back of the mouse with a total volume of 200 μL. In the MN group, the patch was pressed against the skin on the back of the mouse with fingers for 10–15 min. After removal of the patch, any remaining liquid on the skin surface was wiped off with a cotton ball. Fluorescence signals were collected using an IVIS Spectrum (PerkinElmer, Waltham, MA, USA) with excitation light set at 640 nm and emission light at 700 nm. The exposure time was 5 s, and fluorescence changes were observed.

### 5.9. PLGA-DDAB-rETX^Y196E^-C dMN Immunization in Mice

SPF-grade female Balb/c mice aged 6–8 weeks were obtained from Beijing Sibeifu (Beijing, China) Biotechnology Co., Ltd. The mice were randomly divided into eight groups, with five mice in each group (see Table 2 for details). The immunization schedule is shown in Figure 5A, with immunizations administered on days 0, 14, and 28. The immunization dose and route were consistent for each immunization. Blood samples were collected from the tail vein of each mouse one week after each immunization to measure antibody titers.

### 5.10. Mouse Challenge Experiment

Natural ETX toxin challenge experiments were performed 14 days after the third immunization in mice. Mice were injected with 500 μL of 100 × LD_50_ of toxin via intraperitoneal injection. The survival status of each group of mice was recorded for 3 consecutive days.

### 5.11. ELISA Detection of Antibody Levels in Immune Sera and the Content of MN Vaccine

The titers of ETX-specific antibodies IgG (Solarbio C1055, Beijing, China), IgG1 (ab97240, Abcam, UK), and IgG2a (ab97245, Abcam, UK) were measured by indirect ELISA. In brief, rETX^Y196E^-C protein was diluted to 10 μg/mL with Elisa coating buffer (diluted to 1×, Solarbio C1055, Beijing, China) and added to 96-well ELISA plates (Thermo 442404, Waltham, MA, USA) at 100 μL per well, and incubated overnight at 4 °C. After washing with PBST three times, 5% BSA was added and the plate incubated at 37 °C for 1 h. After washing with PBST three times, mouse sera were diluted with 5% BSA in concentration gradients of 10^−1^, 10^−2^, 10^−3^, 10^−4^, 10^−5^, and 10^−6^, added to ELISA plates at 100 μL per well, and incubated at 37 °C for 1 h. After washing with PBST three times, horseradish peroxidase-labeled goat anti-mouse secondary antibodies IgG (1:5000 dilution), IgG1 (1:10,000 dilution), and IgG2a (1:10,000 dilution) were added to each well and wells were incubated at 37 °C for 1 h. After washing with PBST three times, TMB colorimetric substrate (Solarbio PR1210, Beijing, China) was added at 100 μL per well, and plates were then incubated for 10 min in the dark. Finally, the reaction was stopped by adding 50 μL of ELISA stop solution (Solarbio C1058, Beijing, China) to each well, and the OD_450_ values were immediately measured. The obtained OD values were corrected for the background interference using the blank control OD value, and immune serum samples with OD values greater than 2.1 times the negative control OD value were considered positive. The MN rETX^Y196E^-C recombinant protein content was determined using the sandwich ELISA method established in our laboratory, detailed as described in reference [38].

### 5.12. In Vitro Neutralization Assay

In vitro neutralization animal toxicity assay. Female Balb/c mice at 6–8 weeks of age were selected for the experiment, with five mice in each group. Mouse sera were obtained after triple immunization and diluted 20-fold. Equal volumes of each diluted serum and 10 × LD_50_ GST-ETX were mixed and incubated at 37 °C for 30 min. The mixture was then intraperitoneally injected into each mouse at a volume of 500 μL and the survival status of the mice in each group was observed.

In vitro neutralization cytotoxicity assay. The CT_50_ of GST-ETX on MDCK cells was measured using the MTS method. E-plate wells were filled with 50 μL of DMEM culture medium containing 10% FBS and placed on the detection platform of the instrument to measure the baseline for 50 min. The MDCK cells with good growth status were resuspended with DMEM culture medium and counted. The cell concentration was adjusted to 2 × 10^6^ mL^−1^ with DMEM culture medium and plated in the E-plate at 50 μL per well. Mouse serum obtained from a mouse in the immunized group was diluted 20-fold, and GST-ETX was diluted to 20 × CT_50_, with equal volumes mixed and incubated at 37 °C for 30 min. Next, 150 μL of the mixture was added to each well, and duplicate wells were prepared for each concentration. The E-plate was placed on the detection platform, and after incubating at 37 °C for 10 min, the instrument was run. After 72 h, the E-plate was removed and the data were analyzed using the RTCA Software 2.1.5.

### 5.13. Cytokine Detection of Splenocytes Stimulated by rETX^Y196E^-C In Vitro

Splenocyte cytokine flow cytometric analysis was performed on mice that were challenged with the toxin for 7 days, with three mice in each group. Immediately after cervical dislocation, the spleen was removed and minced in a biosafety cabinet, and then gently ground and suspended in 1640 medium containing 10% FBS, 0.1 mM 2-mercaptoethanol, 100 U/mL penicillin, and 100 U/mL streptomycin to make a single cell suspension. Four million cells were plated in a 12-well plate, and to each well was added 2 μg anti-mouse CD28 antibody (553294, BD Biosciences, San Jose, CA, USA), 2 μg anti-mouse CD49d antibody (553153, BD Biosciences, USA), and 40 μg rETX^Y196E^-C (positive control added 20 ng anti-mouse CD3 antibody (100339, BD Biosciences, USA), negative control added only 2 μg anti-mouse CD28 antibody and 2 μg anti-mouse CD49d antibody), as well as 1 µg CD154 BV605 (745242, MR1, BD Biosciences, USA) and 1 µg CD107a BV786 (564349, 1D4B, BD Biosciences, USA), and the appropriate amount of culture medium was added to a total volume of 2 mL. The cell culture plate was placed in a CO_2_ incubator at 37 °C with 5% CO_2_ for 18 h. Two microliters of BFA (555029, BD Biosciences, USA) and 1.4 μL of monensin (554724, BD Biosciences, USA) were added per milliliter of medium, and the cells were further incubated at 37 °C and 5% CO_2_ for another 6 h. The cells were transferred to flow tubes (352054, BD Falcon, USA), and the following reagents were added in sequence: receptor blocking reagent (553141, BD Biosciences, USA); surface staining antibody mix (CD3 PerCP-eFluor710 (46-0032-82, 17A2, Invitrogen, Waltham, MA, USA); CD4 R718 (553046, RM4-5, BD, USA); CD8 BV510 (100752, 53-6.7, Biolegend, San Diego, CA, USA); wash buffer (DPBS containing 1% FBS); live/dead dye (423105, Biolegend, USA); fixation solution; permeabilization washing buffer (554714, BD Biosciences, USA); intracellular staining antibody mix (IL-2 APC (503810, JES6-5H4, Biolegend, USA); IL4 BV421 (504120, 11B11, Biolegend, USA); IL-6 PE (554401, MP5-20F3, BD Biosciences, USA); IFN-gamma Alexa Fluor 488 (505813, XMG1.2, Biolegend, USA); wash buffer; and PBS. Data were analyzed using flow cytometry (Aurora, Cytek, Fremont, CA, USA) and SpectroFlo software (Cytek, V2.2.0.4).

### 5.14. Pathological Damage in rETX^Y196E^-C Immunized Mice

The brain and kidneys of mice were taken to assess pathological damage after the toxin challenge. The positive control group mice were injected intraperitoneally with 100 × LD_50_ GST-ETX, and the negative control group consisted of newly purchased mice. The brain and kidneys of the mice were taken under sterile conditions, the adipose tissue carefully removed, and then washed with PBS to remove blood. Organs were then fixed in 4% paraformaldehyde (P1110, Solarbio, Beijing, China) for 24 h, embedded, stained with hematoxylin and eosin (HE), and observed using slicing.

### 5.15. Statistical Analysis

GraphPad Prism 8.3.0 was used for statistical analysis. All experiments were repeated at least three times, and data are expressed as means ± standard deviation. Mouse sera antibody titers were analyzed with a two-way ANOVA; statistically significant differences were defined as *p* < 0.05.

## Figures and Tables

**Figure 1 toxins-15-00461-f001:**
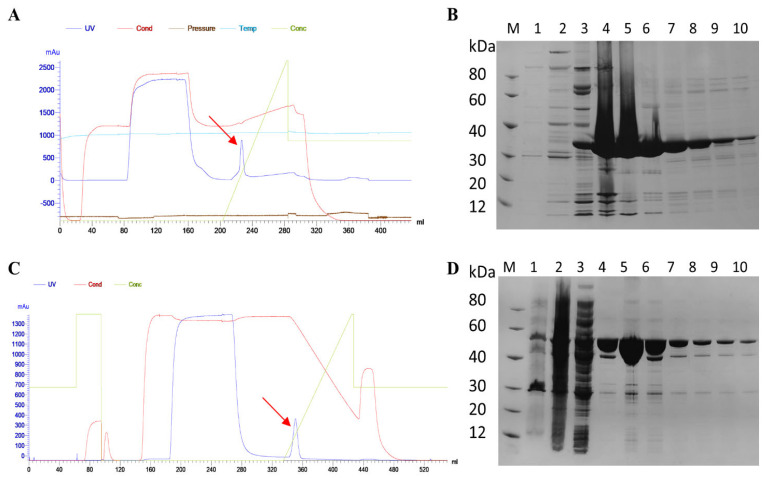
(**A**) Purification profile of rETX^Y196E^-C protein, with the red arrow indicating the target protein peak. (**B**) SDS-PAGE analysis of rETX^Y196E^-C protein. M: molecular weight marker; lanes 1–2: induced bacterial culture, sonicated pellet and supernatant; lane 3: flow-through; lanes 4–10: target protein. (**C**) Purification profile of GST-ETX protein, with the red arrow indicating the target protein peak. (**D**) SDS-PAGE analysis of GST-ETX protein. M: molecular weight marker; lanes 1–2: induced bacterial culture, sonicated pellet and supernatant; lane 3: flow-through; lanes 4–10: target protein.

**Figure 2 toxins-15-00461-f002:**
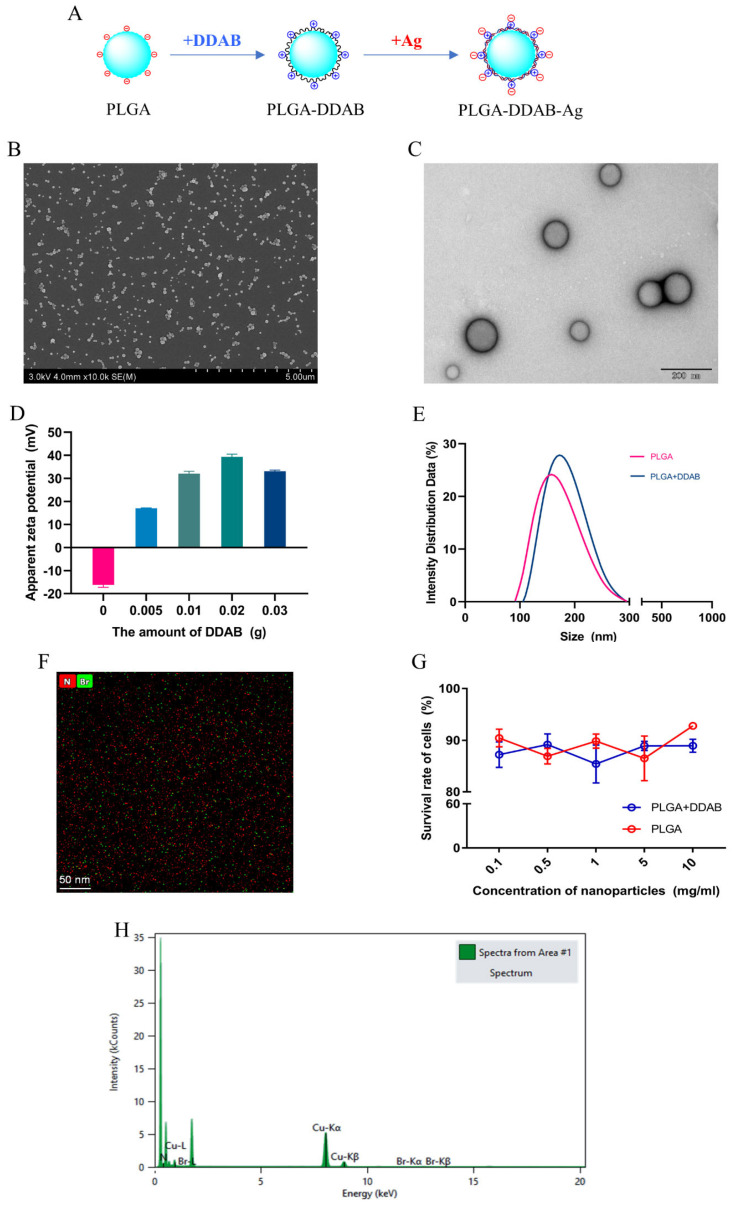
Characterization of nanoparticles. (**A**) Schematic diagram of PLGA NPs with the introduction of cationic lipid DDAB and antigen (Ag). (**B**) Scanning electron microscopy image of PLGA-DDAB NPs (scale bar: 5 μm). (**C**) Field emission transmission electron microscopy image of PLGA-DDAB NPs (scale bar: 200 nm). (**D**) Effect of the amount of DDAB on the surface ZETA potential of PLGA NPs. (**E**) Comparison of particle size and distribution between PLGA and PLGA-DDAB NPs. (**F**) Testing images of Br and N elements in PLGA-DDAB NPs. (**G**) Cell viability of MDCK cells after co-incubation with different concentrations of PLGA and PLGA-DDAB NPs. (**H**) EDS spectrum of PLGA-DDAB NPs.

**Figure 3 toxins-15-00461-f003:**
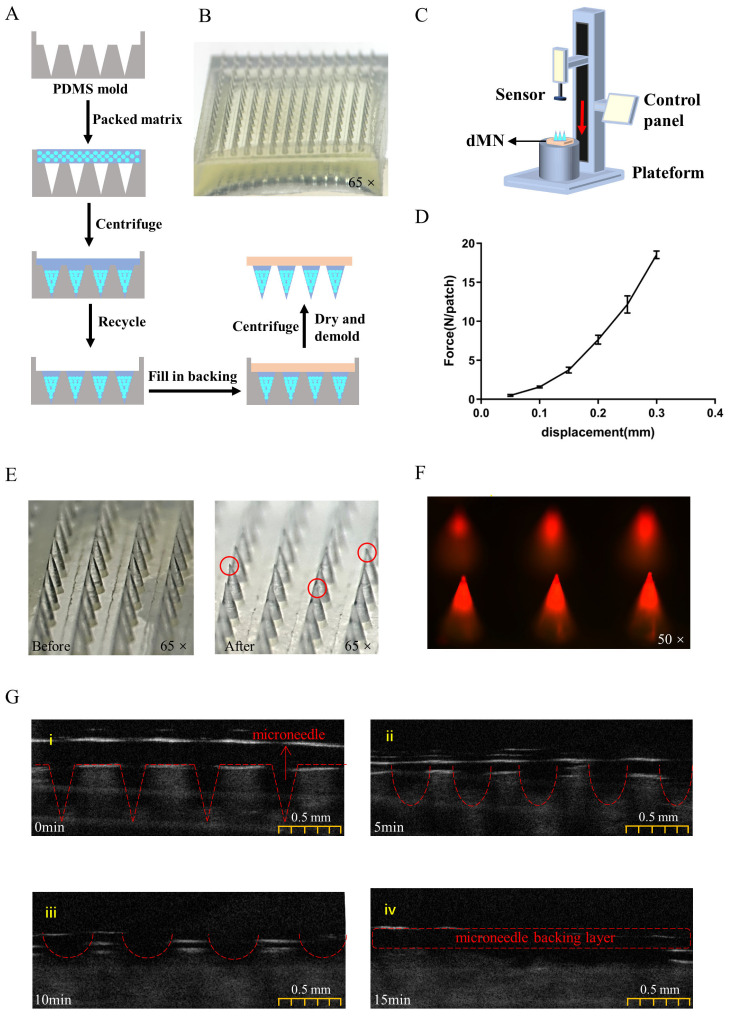
Preparation and evaluation of PLGA-DDAB dMNs. (**A**) Schematic diagram of PLGA-DDAB dMN preparation process. (**B**) Body mode microscope image of dMN. (**C**) Schematic diagram of dMN force-displacement testing. (**D**) Force-displacement curve of PLGA-DDAB dMN. (**E**) Microscopic images of dMN before and after compression, with red circles indicating needle tip deformation. (**F**) Fluorescence image of PLGA-DDAB dMN encapsulating Rhodamine B. (**G**) Optical coherence tomography (OCT) image of PLGA-DDAB dMN dissolved after in vitro skin insertion (scale bar is 0.5 mm). (**i**): 0 min, (**ii**): 5 min, (**iii**): 10 min, (**iv**): 15 min images after the microneedle was inserted into the skin.

**Figure 4 toxins-15-00461-f004:**
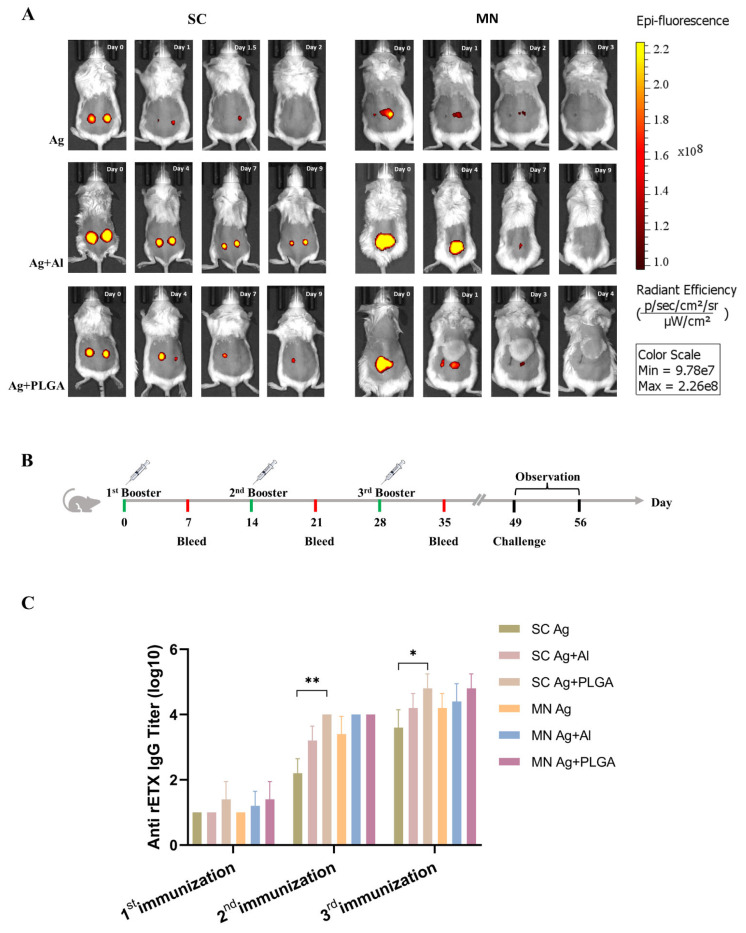
(**A**) Comparison of in vivo release time of rETX^Y196E^-C protein in mice. The relative residence times of the antigen at the injection site for the subcutaneous immunization group and the MN group are Ag + Al > Ag + PLGA > Ag. (**B**) Timeline of the immunization process. (**C**) Antibody titers in the serum of mice from the six immunization groups after three immunizations, * *p* < 0.05, ** *p* < 0.01. MN: microneedle patch; SC: subcutaneous injection.

**Figure 5 toxins-15-00461-f005:**
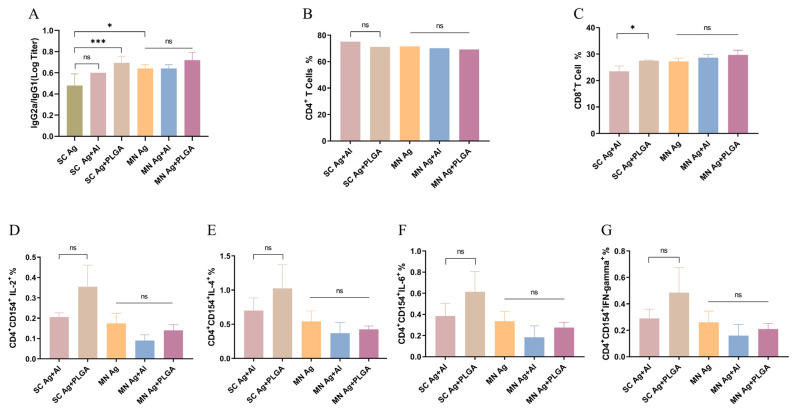
(**A**) IgG2a/IgG1 ratio of specific antibodies in mouse serum on day 35. (**B**) Percent CD4^+^ T cells. (**C**) Percent CD8^+^ T cells. (**D**) Percent CD4^+^CD154^+^ IL-2^+^. (**E**) Percent CD4^+^CD154^+^IL-4^+^. (**F**) Percent CD4^+^CD154^+^IL-6^+^. (**G**) Percent CD4^+^CD154^+^IFN-gamma^+^. * *p* < 0.05, *** *p* < 0.001. MN: microneedle patch; SC: subcutaneous injection.

**Figure 6 toxins-15-00461-f006:**
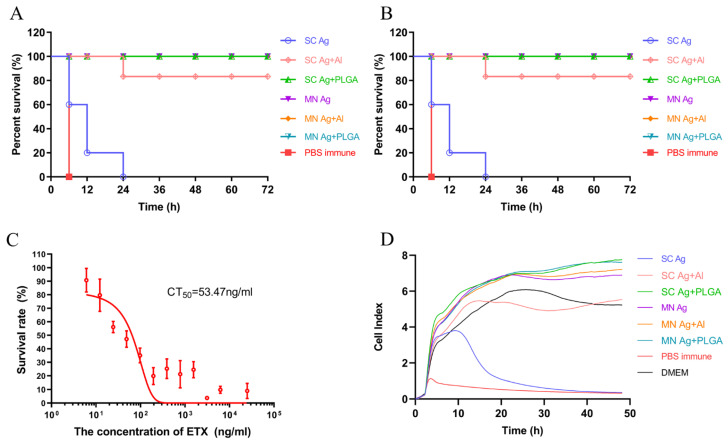
(**A**) Survival curve of mice after toxin challenge. (**B**) In vitro neutralization assay of animal toxicity. (**C**) CT_50_ of GST-ETX on MDCK cells. (**D**) In vitro neutralization assay of cell toxicity: cell proliferation curve. (**E**) Pathology of mouse kidney and brain after toxin challenge. The positive control group exhibited kidney bleeding and edema (red arrow), brain tissue vacuolization (red arrow), and edema (black arrow). No pathological injury was observed in any of the immunized groups or the PBS negative control group. MN: microneedle patch; SC: subcutaneous injection.

**Table 1 toxins-15-00461-t001:** Introducing DDAB resulted in changes in the mean ± standard error of size, zeta potential, and PDI of PLGA nanoparticles.

Group	Size (nm)	Zeta Potential (mv)	PDI
PLGA	164.0 ± 37.4	−16.2 ± 1.0	0.157 ± 0.016
PLGA-DDAB	176.7 ± 33.4	39.3 ± 1.2	0.052 ± 0.022

**Table 2 toxins-15-00461-t002:** Vaccination experiment groups.

No.	Group (*n* = 5)		Dose of Antigen (µg)	NPs Content (µg)
1	MN	Ag	10	-
2		Ag + Al	10	-
3		Ag + PLGA	10	400
4		PBS	-	-
5	SC	Ag	10	-
6		Ag + Al	10	-
7		Ag + PLGA	10	400
8		PBS	-	-

MN: microneedle patch; SC: subcutaneous injection.

## Data Availability

The data presented in this study are available on request from the corresponding author.

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
