# Peer review of "PLGA Nanoparticle-Based Dissolving Microneedle Vaccine of Clostridium perfringens ε Toxin"

_toxins, 2023, doi:10.3390/toxins15070461_

Round 1
Reviewer 1 Report
The current study is novel in various aspects. The authors have attempted a fusion of new strategies for generating a vaccine candidate for Epsilon toxin. However, I feel the manuscript needs to address following comments to make it more scientifically engaging.
Major comments:
Many Images in this manuscript are unclear. It becomes difficult to make any observations and conclusions. Fig 1A, 1C, 2F, 2G, 6E need to be revised thoroughly. The resolution is bothering.
Line 94: What is the effect of mutation in the toxin sequence? Its mentioned that the attenuated form was used for mice immunizations. However, there is no further explanation about this particular mutation.
Minor comments:
Line 44-49: Reference missing
Line 79: Instead of ‘Xiaoling Zheng et al’ write ‘Zheng and co-workers’
Fig 5A: Statistics missing
Minor editing of English language required
Reviewer 2 Report
This manuscript describes extensive study on development of C. perfringens epsilon toxin using a novel approach, covering preparation of vaccine, administration, and immune response. Although manuscript is generally well written, it contains many unique abbreviations, which cause somewhat difficulty in reading. This reviewer suggests following revisions to improve manuscript.
1. Abstract: Words of full spelling are necessary when they appear first in text, e.g., "NPs" and "MN", although that of "dMN" is shown.
2. Abstract: Add "," to separate sentence between 100% and while.
3. Abstract and main text : "Aluminum" should add to "Al" when it first appear in text.
4. Section 2.2, line 2-3: full words of DDBA is already described in Introduction. Only DDBA should be written.
5. Figure 2A and main text : "Ag" should be explained in text and also figure legend. According context, readers can understand it as antigen, but it also looks like an element symbol of silver.
6. Figure 2F: It shows almost black field and image can be hardly seen. Change contrast slightly for readers to understand this image.
7. Figure 3G: Meaning of these figures were not understandable, even there are descriptions in figure legend. Please explain more detail for readers unfamiliar with this experiment to understand.
8. Figure 4C, Figure 5, 6: "SC" is shown, but this full word (subcutaneous injection) is not found. It should be added to text and figure legend when it appears first. (it is shown in Table 2 footnote.)
Round 2
Reviewer 1 Report
I thank the authors for addressing majority of the comments to my satisfaction. However, the authors need to seriously work more on Fig. 6E, even if it means re-imaging these sections. The authors should either use a higher resolution microscope or seek help from a trained personnel for this data representation. Every image in this panel, for mice tissue, appears as a big pink patch. Even after zooming out by 175-200%, I could not get any clear view of the individual nuclei or other structural details. A random 'google images' search for mice kidney and brain histology shows such detailed images. The images in the manuscript are in no way defining histological resolutions. It would be difficult for the readers to understand and conclude from these images if they are not clear enough.
Author Response
Manuscript ID: toxins-2495675
Title: PLGA nanoparticle-based dissolving microneedle vaccine of Clostridium perfringens ε toxin
Dear editor and reviewers,
Thank you once again for dedicating your time and effort to review our manuscript. We genuinely appreciate the positive comments and valuable suggestions you have provided which helping us to improve the quality of our image. we have thoroughly revised image Fig 6E, resulting in an improved resolution and enabling readers to clearly visualize individual nuclei and other structural details. Thank you for bringing these areas of improvement to our attention.
The modified manuscript has been uploaded as an attachment. We would like to express our sincere gratitude once again for all of your valuable and thoughtful comments.
